# Emotional and Cognitive Aptitudes and Successful Academic Performance: Using the ECCT

**DOI:** 10.3390/ijerph182413184

**Published:** 2021-12-14

**Authors:** María Vera, José A. Cortés

**Affiliations:** 1Social Psychology Department, Universidad Pablo de Olavide, 41013 Sevilla, Spain; 2Instituto Tecnológico Superior Cordillera, Quito 170104, Ecuador; jose.cortes@cordillera.edu.ec

**Keywords:** cognitive, emotional, academic performance

## Abstract

Understanding factors that influence academic performances is vital. The aim of this study is to longitudinally test, with three timepoints, the unique contribution of several predictors to academic performance. In a sample of 796 Ecuadorian students, dominance analyses were performed with the R program to test the relative and unique importance of the seven variables under study (verbal aptitude, numerical aptitude, abstract reasoning, emotional regulation scenarios, emotional regulation self-questionnaire, and academic performance measured in timepoint one and two) for academic performance, measured in timepoint three in the entire sample and separately in each of the ten degrees in the academic center. Results show that the strongest predictors are past academic performance, followed by gender, numerical aptitude, scenarios, verbal aptitude, abstract reasoning, and, finally, the emotional regulation self-questionnaire. This study contributes to explaining the complex topic of academic performance. More studies are needed in order to better understand the role played by emotional intelligence, as well as differences between different degrees or areas of study.

## 1. Introduction

Education is a key aspect of the development of any country. Education is basic and essential for future development; hence, it is a primary aspect on the agendas of every country. Education is the new priority, which contributes to improving the competitiveness of national economies in the context of increasing globalization, since the future and positioning of any country depends on its human capital and, therefore, on citizens’ skills and know-how [1].

As with any other matter, if you cannot measure it, you cannot improve it. Therefore, academic performance becomes vital when we deal with educational issues. Academic performance becomes important since it is commonly used as a reflection of education. In the past century, academic performance has become the gatekeeper to institutions of higher education, shaping career paths and individual life trajectories. As the future of any country depends on the quality of its professionals, it is vitally important for all societies to have excellent students who become successful professionals in their fields [2]. In our system composed of qualifications and merits, academic performance is one of the few scores available to measure success.

There is no common agreement regarding the evaluation of academic performance, but measures of cognitive skills or declarative knowledge are the main factors evaluated [3]. Moreover, the most commonly used indicator to measure academic performance is usually the Grade Point Average (GPA). In this study, we measure the GPA of the first three semesters of students included in the sample.

Considering the importance of academic performance seems crucial to understand its predictors. For this reason, research has focused on identifying predictors of academic performance, with intelligence and effort emerging as core determinants. In this vein, due to the importance of academic performance in higher education, it is vital to understand the factors that influence it [4].

Cognitive ability was identified as having central importance in predicting academic outcomes [5,6], and cognitive ability is a strong predictor of academic performance [7]. For more than a century, psychologists and educational theorists have been interested in the links between various tests of mental ability and academic performance [8]. Despite the development and review of research ideas during this time, a crucial factor in predicting academic achievement continues to be an individual’s level of general cognitive ability.

In the present paper, we will show some factors that may influence academic performance. The next part of the introduction will be divided into three sections, one for each academic performance predictor. In each one, every predictor will be explained. Their relationships with academic performance and the hypotheses will close each section: firstly, cognitive aptitudes, in which verbal aptitude, numerical aptitude, and abstract reasoning will be analysed; secondly, past performance, where students´ academic performance in their first and second semesters will be analysed; and thirdly, emotional aptitudes, in which rating scales and ability scales will be differentiated, and the effect of each of them on academic performance will be analysed.

Therefore, in the present paper, several variables will be tested as predictors of academic performance, specifically, cognitive and emotional aptitudes and past academic performance. Thus, the main objective is to longitudinally test, through three semesters, the unique contribution of several predictors to academic performance using dominance analyses.

### 1.1. Cognitive Aptitudes

Cognitive aptitudes are not new in the literature (see [9]) or obsolete [10]. As mentioned above, they are strongly related to academic performance. In this study, we take verbal aptitude, numerical aptitude, and abstract reasoning into account. Verbal aptitude is the ability to understand the meaning of words and to use them effectively; the ability to understand language, understand the relationships between words, and understand the meaning of complete sentences and paragraphs [11]. Therefore, this skill includes the ability to quickly identify critical information and draw logical conclusions from written facts or data. Numerical aptitude is the ability to understand number relationships, use numbers effectively, and understand quantitative material [11]. Finally, abstract reasoning refers to the ability to make and interpret drawings or figures, establishing relationships and formulating equivalences by continuing series or eliminating figures.

Therefore, we expect cognitive aptitudes to be related to future academic performance. Therefore, the following hypotheses were considered:

**Hypothesis** **1** **(H1).***Cognitive aptitudes explain unique variance in future academic performance*.

**Hypothesis** **1** **(H1a).***Verbal aptitude explains unique variance in future academic performance*.

**Hypothesis** **1** **(H1b).***Numerical aptitude explains unique variance in future academic performance*.

**Hypothesis** **1** **(H1c).***Abstract reasoning explains unique variance in future academic performance*.

### 1.2. Past Performance

‘‘Success breeds success’’ summarizes the idea that the best predictor of future performance is past performance [12]. In this vein, in a literature review on predicting academic success showed that prior academic achievement and student demographics were present in 69% of the research papers [13]. According to the authors, this observation was aligned with the results of a literature review emphasizing that the GPA is the most common factor used to predict future student performance. Several studies confirm that pre-university data include high school results that help to understand the consistency in students’ performance [14,15,16]. In this vein, despite differences in course content and grading criteria, the high school grade point average (GPA) is a stronger predictor of the university GPA than the scholastic aptitude test (SAT), the most widely used, standardized, college admissions test in North America) [17]. Therefore, we expect:

**Hypothesis** **2** **(H2).***Past academic performance explains unique variance in future academic performance*.

**Hypothesis** **2** **(H2a).***GPA in Time 1 explains unique variance in GPA in Time 3*.

**Hypothesis** **2** **(H2b).***GPA in Time 2 explains unique variance in GPA in Time 3*.

### 1.3. Emotional Aptitudes

Although past achievement and cognitive ability were tested as the most strongly related predictors of academic performance, there were several attempts to explain, and thus try to predict academic performance. Therefore, the factors contributing to one individual’s achievement compared to another’s in educational settings are the topic of extensive debate and continue to attract investigative interest [18]. Distinct strands of evidence indicate that predictions of academic performance may be more accurate if they are based on the assessment of a variety of individual differences, not simply on past achievement and cognitive capacity [17]. For instance, intelligence, personality, and interests [19], the Five Factor personality traits [20], motivation, self-regulatory learning strategies, and learning styles were also found to predict academic performance when controlling for the effects of intelligence and personality [21]. Moreover, the relationship between conscientiousness and academic performance was largely independent of intelligence [17]. In this vein, although intelligence accounted for the greatest amount of variance, the combined effects of curiosity and effort equaled the impact of intelligence on academic performance [2]. Moreover, academic self-efficacy was also shown to predict academic performance [4]. In fact, tree cover near schools, green window views, college preparatory exams, and end-of-semester grades are the most promising indicators of a beneficial link between school green space and academic performance [22].

As we can see, there is great interest and, therefore, extensive literature on academic performance predictors. There are many studies and many variables that were found to be related to academic performance. They are not contradictory studies, but rather pieces of a complex puzzle that we still do not understand in its entirety. Recently, in education, there is a growing consensus among educators, researchers, and policymakers that emotional intelligence is an important skill for students to develop, both for their future wellbeing and their future workplace success [23]. According to [24], students with high global trait emotional intelligence have superior emotional information processing skills, regulation, and coping skills, and they may be more successful in coping with the demands of school and the peer context. Thus, it seems that emotional intelligence may be a crucial part of the complex puzzle.

Emotional intelligence is a well-known construct in psychology, although it is not as well-established as cognitive abilities. It first appeared in 1990 [25], but the concept was relatively unknown until it was popularized by science journalist Daniel Goleman in 1995 in his book Emotional Intelligence: Why It Can Matter More Than IQ. Since then, the most well-known model has become the hierarchical four-branch model [26]. This model outlines four key branches of emotion-related abilities that range in complexity from low-level information processing to the strategic and deliberative use of emotional information to meet personal goals. These four branches are: (a) perceiving emotions accurately, (b) using emotions to facilitate decision making, (c) understanding emotions, and (d) managing emotions to upregulate positive emotions and downregulate negative emotions. 

In this study, we focus on emotion regulation, the ability to manage emotions in oneself and others to achieve a desired outcome such as personal growth [27]. Given that emotion regulation is the last step in the model and the one that was shown to have the strongest relationship with academic performance [28], we decided to include it in this study as a measure of emotional aptitude to predict future academic performance.

As stated above, emotional intelligence is a well-known construct, and although the more recognized model is the hierarchical four-branch model, it has several conceptualizations. These can be broadly bifurcated by the type of measurement technique used [23]. Ability scales require test-takers to demonstrate knowledge or process emotion-related information to provide a response. Rating scales require test-takers to rate their agreement with a series of statements about themselves. On the one hand, ability scales involve processing and manipulating emotional information, defined as the ability to perceive, use, understand, and manage emotions. This ability is measured with objective test items, such as asking test-takers to identify the emotion in a facial expression orto judge how effective an action would be in managing an emotional situation [29]. On the other hand, rating scales are based on self-perceptions. It was suggested that trait emotional intelligence should be conceptualized as a lower-order personality construct that captures variance not accounted for by existing personality measures [30].

Current evidence suggests that rating scales and ability scales of emotional intelligence capture different constructs and are only weakly related to each other [23], for instance, meta-analytic correlations ranging from 0.12 to 0.26 [31]. Moreover, different meta-analyses show that both the abilities of emotional intelligence and trait emotional intelligence are associated with academic performance, although ability scales seem to show a stronger relationship with academic performance [29]. Therefore, in this study, we measured emotion regulation through both types of scales: ability and trait. Thus, we can find out what the relationship is between them and with academic performance separately.

There is a large body of literature that links emotional intelligence with academic performance. The meta-analysis by [23] summarizes three meta-analyses carried out previously, all of which found a positive association. First, with a corrected correlation of 0.10 between emotional intelligence and academic performance [32]. Second, regarding emotional intelligence rating scales, a corrected correlation of 0.20 with academic performance [33]. Third, a relationship between emotional intelligence and academic performance (0.17) [17]. The authors obtained a significant positive correlation between overall emotional intelligence and academic performance (0.20, 95% CI (0.17, 0.22)). In addition, emotional intelligence (trait or emotional self-efficacy) has implications for academic performance, with effects mainly relevant to groups with a lower cognitive ability [34]. 

Therefore, there seems to be a link between emotional intelligence and academic performance. Moreover, the main predictor of academic performance seems to be emotion regulation. Furthermore, there are roughly two kinds of scales that measure emotional intelligence: rating scales and ability scales. They seemingly capture different constructs of emotional intelligence and are only weakly related to each other. By collating all of this information together, we expect that:

**Hypothesis** **3** **(H3).***Emotion regulation explains unique variance in future academic performance*.

**Hypothesis** **3** **(H3a).***Ability scales (scenarios) explain unique variance in future academic performance*.

**Hypothesis** **3** **(H3b).***Trait scales (self-questionnaire) explain unique variance in future academic performance*.

**Hypothesis** **3** **(H3c).***There is a low relationship between ability and trait scales*. 

**Hypothesis** **3** **(H3d).***Ability scales explain stronger future academic performance compared to trait scales*.

In summary, this paper will help to understand the unique contribution of different factors (i.e., cognitive aptitudes, past academic performance and emotion regulation) in academic performance in a longitudinal study with three timepoints (see Figure 1).

## 2. Materials and Methods

### 2.1. Procedure and Sample

This study was carried out in a Technological Institute in Ecuador; therefore, it included a convenience sample. This institute has a mean of 5200 enrolled students and offers a total of ten degrees. With the aim of selecting those students who can achieve the most from their studies, the ECCT [35] admission test is given every semester (twice a year). In 2018 (first semester), 1750 students performed the ECCT, and 1338 managed to enroll in the institute and start their studies. From that moment, these students were registered and coded, including a record of their grades during the following three semesters. Throughout the process, the confidentiality of the data was guaranteed because we worked with codes, and the students agreed to share their grades and ECCT results with the research office, ensuring total anonymity of the data and their exclusive use for research tasks.

Of these 1338 selected students who finished the first semester, 1028 finished the second semester, and 796 finished the third. These dropout figures are typical of the country and the educational level. Thus, the final sample was composed of 796 students, 60.9% women and 39.1% men with a mean age of 21 years old (SD = 4.3). Regarding their degrees, 19.3% studied banking and financial administration, 18.7% child talent development, 14.7% human resources management, 10.1% graphic design, 8.5% tourism and hospitality management, 8% systems analysis, 6.4% industrial and production management, 5.3% apothecary and pharmacy management, 4.5% optometry, and 4.4% marketing.

### 2.2. Instruments

Cognitive aptitudes. The Emotional and Cognitive Competence Test (ECCT, [35]) was used to measure several variables: (1) Verbal aptitude. As [11] defined, this is the ability to understand the meaning of words and use them effectively; the ability to understand language, the relationships between words, and the meaning of complete sentences and paragraphs. Thus, this aptitude is measured through the comprehensive reading of texts (6 items), synonyms (6 items), antonyms (6 items), verbal analogy (6 items), and syllogisms (6 items). (2) Numerical aptitude is the ability to understand number relationships, use numbers effectively, and understand quantitative material [10]. This aptitude is measured through a series of numbers (10 items), arithmetic calculations (10 items), and mathematical problems (10 items). (3) Abstract reasoning is the ability to make and interpret drawings or figures, establishing relationships and formulating equivalences by continuing series or eliminating figures. It is measured with two types of exercises: progressive series (14 items) and topology (16 items). Due to the length of the examples, they were not included in this section, but can be viewed in the manual [34]. All items are treated as dichotomous variables because we measured success or error, given that there is only one correct answer. The final score for each aptitude is the sum of the correct answers. Thus, the maximum score of cognitive aptitudes is 90 (30 for verbal aptitude, 30 for numerical aptitude, and 30 for abstract reasoning), and the minimum score is 0.

Emotional aptitudes. The Emotional and Cognitive Competence Test (ECCT, [35]) was used to measure emotional regulation. It is the last of the four dimensions of emotional intelligence, understood as the ability to recognize, use, understand, and manage one’s own and others’ emotional states to resolve problems and regulate behavior [26]. Emotional regulation was measured through: (1) Scenarios, using a total of six scenarios, three for managing one’s own emotions and three for managing others’ emotions. Each of these scenarios had 5 possibilities for action, and the student had to decide which was more or less effective in solving the proposed problem. The student received a score for each scenario ranging from 0 (when incorrectly evaluating the 5 action strategies) to 20 (when effectively evaluating the 5 action strategies). The final score was the mean of the 6 scenarios; thus, scores ranged from 0 to 20. (2) Questionnaire. The ECCT also had a self-report measure composed of 15 items where students had to indicate to what extent they agreed or disagreed with each of the proposed statements. The answers are Likert type with 5 response options from 1 (Strongly disagree) to 5 (Strongly agree) (i.e., “I get carried away easily by anger”).

Academic performance. Each degree lasted six semesters. For this research, one mean was calculated for each of the first three semesters. Thus, for each student in any of the ten possible degrees, we calculated three academic performance scores during this time. The first score was the mean for all subjects taken in the first semester, the second score was the mean for all subjects taken in the second semester, and the third score was the mean for all subjects taken in the third semester. The subjects were different depending on the degree. Scores ranged from 0 to 10.

### 2.3. Data Analyses

First of all, Kolmogorov–Smirnov test was performed in order to test normality. Since significant results were obtained, the normal distribution of the study variables could not be assured. Descriptive analyses were performed on the variables. Means, standard deviations, internal consistencies, and Spearman correlations among the study variables were calculated using the IBM-SPSS 26.0 program. Regarding internal consistencies, in the case of cognitive aptitudes, the Kuder–Richardson Formula 20 (KR20; [36]) index was used. As [37] stated, KR20 is a well-known measure in classical test theory, and it is widely used to evaluate the internal consistency of cognitive and personality tests. The formula for the computation of KR20 is suitable for items with dichotomous scores [38]. Values < 0.5 were low, 0.5–0.6 moderate, 0.6–0.7 good, 0.7–0.8 high, and >0.8 very high [39]. Moreover, in the case of scenarios and the self-questionnaire, Cronbach’s alpha was used. In these cases, values > 0.70 showed a good internal consistency [40]. Moreover, Extracted Mean Variance (AVE) and Composite Reliability (CR) were calculated. For AVE, acceptable value is greater than 0.50 [41]. Additionally, regarding CR, values greater than 0.70 were acceptable [42]. Moreover, in order to assess the validity of the constructs of the ECCT, a Confirmatory Factor Analysis (CFA) was performed using the AMOS software package [43]. The goodness-of-fit of the models was evaluated using absolute and relative indexes. The three absolute goodness-of-fit indexes that were calculated were: (1) the χ^2^ goodness-of-fit statistic; (2) the Goodness-of-Fit Index (GFI); and (3) the Root Mean Square Error of Approximation (RMSEA). Additionally, we computed a relative index: Comparative Fit Index (CFI). Because the distribution of the GFI is unknown, no statistical test or critical value is available [44]. Values below 0.06 for the RMSEA were indicative of an acceptable fit [45], whereas a cut-off value close to 0.95 for CFI was considered to be indicate an adequate model fit [45].

In order to test whether age, gender, and the studied degrees affect the variables under study, Kruskal–Wallis and Mann–Whitney tests were performed. In the case of age, we recodified it into four groups based on the quartiles distribution: the first group (*n* = 253) was under 19 years old; the second group (*n* = 234) was between 19 and 21 years old; the third group (*n* = 133) was between 21 and 23 years old; and the fourth and final group (*n* = 155) was over 23 years old. Twenty-one students decided not to report their age. 

Finally, dominance analyses were performed with the R program to test the relative importance of the seven variables under study (verbal aptitude, numerical aptitude, abstract reasoning, scenarios, emotional regulation self-questionnaire, T1 academic performance, and T2 academic performance) for academic performance in Time 3 in the entire sample and separately in each of the ten degrees in the academic center. Dominance analysis statistic of interest was General dominance, which defines the relevant importance of predictors in a practical and meaningful way [46]. Unlike other dominance analysis statistics, this can almost always be established [47]. In addition, we included gender in the dominance analysis models because it had a significant effect on all the variables under study.

## 3. Results

Regarding descriptive analyses, Table 1 shows the means and standard deviations for all of the variables. As we can see, within cognitive aptitudes, the highest scores are for abstract reasoning and the lowest for verbal aptitude. Almost all internal consistencies are acceptable. There are some exceptions. AVE for verbal aptitude and self-questionnaire are under 0.50, but the other two indexes meet the criteria. Thus, it seems that consistency is not a problem. The CFA yielded the following results: χ^2^_(48)_ = 124.298, GFI = 0.99, RMSEA = 0.029, CFI = 0.98, and GFI = 0.99.

Correlations are as expected. All the cognitive aptitudes have positive and significant relationships. The three academic performance scores have positive and significant relationships. The two ways to measure emotional aptitudes have a positive and significant relationship (ρ = 0.16, *p* = 0.000). Moreover, following [48], these data confirm H3c because this relationship is weak. Regarding the unexpected results, the self-questionnaire measuring emotional regulation has no significant relationship with cognitive aptitudes (only a weak relationship with abstract reasoning). Finally, abstract reasoning has no relationship with academic performance for any timepoint.

Age is not a significant sociodemographic variable because the Kruskal–Wallis test shows no significant effect on any variable under study, except abstract reasoning (χ^2^_(3)_ = 13.53, *p* = 0.004). Specifically, the highest score was in the first group (<19 years old, mean = 23.29), decreasing to become the lowest score in the fourth group (>23 years old, mean = 22.05).

Gender has a significant effect on all the variables under study, except scenarios. Table 2 shows results from the Mann–Whitney test. As Table 2 shows, boys have higher scores for cognitive aptitudes, whereas girls have higher scores for emotional aptitudes and academic performance in the three time periods. Definitively, gender is a variable that must be taken into account.

The degrees that students are studying also seem to be an important variable; the Kruskal–Wallis test shows that it has an effect on all the variables under study, except scenarios. Table 3 shows the results of the Kruskal–Wallis test. The highest verbal aptitude scores are in optometry, and the lowest in banking and financial administration. For numerical aptitude, the highest scores are in systems analysis and the lowest in apothecary and pharmacy management. For abstract reasoning, the highest scores are in systems analysis, and the lowest in human resources management. For the emotional regulation self-questionnaire, the highest scores are in marketing and the lowest in graphic design. For T1 academic performance, the highest scores are in marketing and the lowest in systems analysis. For T2 academic performance, the highest scores are in tourism and hospitality management and the lowest in systems analysis. For T3 academic performance, the highest scores are also in tourism and hospitality management and the lowest in systems analysis. It seems that the analyses may have to take degree that the students study into account.

Finally, Table 4 shows the results of the general dominance analyses for T3 academic performance in the whole sample. As the table reveals, all the study variables explained 25.6% of the variance, and we include the rank that each variable occupies in explaining T3 academic performance by itself, as well as the percentage of what is explained by each variable, taking into account the R^2^ obtained. It is not surprising that the first two positions in the ranking are for academic performance in T1 and T2, with the two past performance measures explaining 82.69% of the R^2^. Thus, H2 (H2a and H2b) is confirmed. The rest (17.31%) of the variance is explained, in order, by: gender (8.85%), we already know that boys have higher scores on cognitive aptitudes, whereas girls have higher scores on emotional aptitudes as well as academic performance at the three timepoints: numerical aptitude (3.08%); scenarios (1.54%); verbal aptitude (1.54%); abstract reasoning (1.15%); and, finally, the emotional regulation self-questionnaire (0.38%). As we can see, each variable under study explains a unique percentage of the variance in T3 academic performance; thus, H1 (H1a, H1b, and H1c) and H3 (H3a and H3b) are confirmed. Moreover, we can also confirm H3d because scenarios (ability scale) explain a higher percentage of the variance than the self-questionnaire (trait scale). After obtaining these data, a question arises: will the same pattern appear in the case of the 10 degrees?

Table 5 shows the results of the general dominance analyses for T3 academic performance in the whole sample and separately by degrees. Several patterns emerge. On the one hand, the variance in T3 academic performance explained by the eight variables under study ranges from 20.3% in the case of graphic design to 68.6% in the case of optometry. T2 academic performance appears as number 1 in the ranking 50% of the time (and 40% of the time as number 2); T1 academic performance appears as number 1 in the ranking 30% of the time (and 40% of the time as number 2); followed by scenarios, which appears as number 1 in the ranking 10% of the time (and 10% of the time as number 2), and numerical aptitude, which appears as number 1 in the ranking 10% of the time (and 0% as number 2). On the other hand, the emotional regulation self-questionnaire appears 40% of the time as number 8 (i.e., the last) in the ranking (and 30% of the time as the penultimate), gender appears as number 8 in the ranking 20% of the time (and 20% of the time as the penultimate measure), scenarios appears as number 8 in the ranking 10% of the time (and 40% as the penultimate measure), abstract reasoning appears as number 8 in the ranking 20% of the time (and 10% of the time as the penultimate measure), and verbal aptitude appears as number 8 in the ranking 10% of the time (and 0% as the penultimate measure). As we can see, there are large differences between the different degrees. Finally, it is important to highlight some points. For instance, in seven of the ten degrees, T1 and T2 academic performance adds up to 50% of the total variance, except in: (a) tourism and hospitality management, in which scenarios and abstract reasoning become important; (b) graphic design, in which numerical aptitude becomes the most important; and (c) optometry, in which scenarios become the most important.

## 4. Discussion

The objective of this paper was to test cognitive and emotional aptitudes and past academic performance as predictors of academic performance. Thus, we have longitudinally shown the unique contribution of several predictors of academic performance using dominance analyses. Specifically, we confirmed that cognitive aptitudes explain a unique percentage of the variance in T3 academic performance (5.77%). This percentage is distributed in 3.08% for numerical aptitude (H1b confirmed); 1.54% for verbal aptitude (H1 a confirmed); and 1.15% for abstract reasoning (H1c confirmed). This pattern was also found when we analyzed the data separately by degrees, with some exceptions. For instance, for tourism and hospitality management and child talent development, abstract reasoning is especially important (19.60% and 12.61%, respectively).

Regarding past performance, we confirmed that T1 and T2 academic performance together explain a unique percentage of the variance in T3 academic performance (82.69%). This percentage is distributed in 36.54% for T1 academic performance (H2a confirmed) and 46.15% for T2 academic performance (H2b confirmed). As we can see, these are the strongest predictors of T3 academic performance, and agree with the idea that ‘‘Success breeds success’’ [12]. Analyzed separately by degrees, as explained above, T1 and T2 academic performance add up to 50% of the total variance in seven of the ten degrees, except for: a) tourism and hospitality management, where scenarios and abstract reasoning add up to 40.53%; b) graphic design, where numerical aptitude is the most important (41.87%); and c) optometry, where scenarios are the most important (45.48%).

We also confirmed that emotional aptitudes explain a unique percentage of the variance in T3 academic performance (1.92%). This percentage is distributed in 1.54% for scenarios, that is, measured by ability scales (H3a confirmed); and 0.038% for the self-questionnaire, that is, measured by trait scales (H3b confirmed). This pattern was also found when we analyzed the data separately by degrees, with some exceptions. For instance, scenarios are important for apothecary and pharmacy management (7.01%), tourism and hospitality management (20.93%), and optometry (45.48%), whereas the highest score for the self-questionnaire was 4.76% in systems analysis. Thus, these percentages seem quite modest, which shows that the understanding and prediction of academic performance is a complex puzzle. Therefore, knowing all of its pieces, no matter how small, is very important. Moreover, we confirmed H3c regarding the relationship between the two ways of measuring emotional intelligence (ability and trait), showing that the relationship between the two measures is low [23]. We also confirmed H3 d because, as other studies revealed, ability scales show a stronger relationship with academic performance than trait scales [29], and emotional self-efficacy (self-beliefs about one’s emotional skills captured by self-rated emotional intelligence) is the least important predictor of academic performance [23].

Finally, regarding sociodemographic variables, we confirmed that age is not significant in predicting academic performance or other variables under study, as other studies have also found [23]. Gender, on the other hand, is significant. First, the results of the Mann–Whitney test show significant differences; boys have higher scores for cognitive aptitudes, whereas girls have higher scores for emotional aptitudes and academic performance in the three times. Second, dominance analyses show that gender explains 8.85% of the variance in T3 academic performance in the whole sample, ranging from 0.43% in child talent development to 21.10% in marketing. Although we obtained higher scores for girls in emotion regulation, samples with larger proportions of females showed weaker effects (of emotional intelligence on academic performance) than more gender-diverse samples [23]. Finally, although we confirmed significant differences in the Kruskal–Wallis test depending on the degree, the pattern found in the general sample is repeated, with a greater or lesser total percentage explained in almost all the degrees, with some exceptions. These exceptions are tourism and hospitality management, in which scenarios and abstract reasoning are the most important; graphic design, in which numerical aptitudes is the most important; and optometry, in which scenarios are the most important. As we can see, our results do not agree with [23], who found that the effects of emotional intelligence on academic performance were stronger in the humanities than in math/science.

### 4.1. Implications

Regarding theoretical implications, due to the importance of academic performance in higher education, understanding the factors that influence it is vital [4]. With this paper, we not only confirmed two classical explanations for academic performance (cognitive aptitude and past performance), but we also confirmed the unique contribution of each of these to future academic performance using the new technique of dominance analyses. Thus, due to these analyses, we can specifically explain what percentage of the variance is explained by each variable. The same thing occurs with emotional aptitudes that explain academic performance. In this vein, we confirmed that ability and trait scales explain different parts of emotion regulation, which opens up a new avenue in this area. Future studies may be interested in answering the following questions: Is it worthwhile to continue to study emotional intelligence through trait scales? Should we analyze other aspects of academic performance, apart from the GPA, so that the trait scale makes more sense? 

Moreover, it is striking that the sample has a high perception of their ability to regulate emotions (average of 4.40 out of 5), whereas in the objective test, they obtain an average of 14.40 out of 20. Therefore, the sample seems to have an overestimated perception of students’ ability to regulate their emotions. Moreover, the objective test (scenarios) has a positive and significant relationship with T3 academic performance, whereas the subjective test (self-questionnaire) does not. Will the same thing occur in other areas? Many of our studies in psychology and education are based on self-perception. Are we relying too much on subjective variables? Should we include more objective variables? Do they really measure two sides of the same coin?

Therefore, we shed some light on the matter by indicating the exact weight of certain variables and pointing to others as possible pieces of the puzzle, such as personality and interests [19], motivation and self-regulatory learning strategies [21], conscientiousness [17], or curiosity [2], all of which are pieces of a complex puzzle with many pieces to discover. In this vein, high school (GPA), university GPA, and the SAT explain the unique variance in GPA, collectively accounting for approximately 25% of the variance and leaving a large amount unexplained [17].

Regarding practical implications, as [49] stated, as the number of higher educational institutions increased, student recruitment and reputation building became more competitive. Although there are many practices that could be used to compete with others, the most promising approach would be to strengthen the institution’s own academic profiling. Thus, creating an admissions test that incorporates the most important predictors of academic performance has vital importance in any country. For instance, the United States has the SAT, Spain has the EBAU (Bachelor Assessment for University Access), and in this study we propose the ECCT for Ecuador.

We know that past academic performance is the strongest predictor of future academic performance, and so we must ensure that students work on their confidence to improve their performance, as well as on their emotional intelligence. For instance, it is tested that social and emotional learning programs increase academic performance [28]. In this vein, a meta-analysis showed that these programs resulted in an 11% improvement in academic performance [50]. Moreover, programs based on the ability model were significantly more effective than those based on mixed models, and understanding emotions showed the largest increase of all the emotional intelligence branches [23]. Thus, programs are effective for increasing emotional intelligence, and particularly its facet of understanding emotions. Therefore, emotional intelligence training seems to produce the strongest increases in the competencies that are most relevant for academic performance. In this vein, it seems important to include not only cognitive but also emotional elements in curricular designs and educational programs.

Finally, it is important to highlight differences between degrees. Although the results show similar patterns, there are also significant differences. Thus, it makes sense to think about differences predictors of academic performance depending on the degree or area. Future studies may try to further explore these issues.

### 4.2. Weaknesses and Strengths

This study has some limitations, although we try to address one of the criticisms of [3], which argued that few studies had been conducted with Spanish-speaking samples. Therefore, more research is needed in Spanish and Latin American populations. We focused only on one Latin American country. Thus, more research is needed in Spanish-speaking samples.

Moreover, as mentioned above, some of the percentages of our variables are very low (e.g., self-perception of emotion regulation). However, this percentage varies depending on the degree, and so it should not be completely ruled out. In this vein, it is important to highlight that we focused only on the fourth branch of emotional intelligence (i.e., emotion regulation), which can be seen as both a weakness and a strength. Although there is ample evidence that emotional intelligence training works, we are not aware of experimental studies on emotional intelligence training that examine the effects of training different branches of emotional intelligence [23]. According to the authors, new studies would isolate the facets of emotional intelligence which were most relevant for improving different outcomes, and they would also provide stronger evidence for the causal direction from emotional intelligence to academic performance.

Finally, some of the strengths that stand out are as follows: (a) this is a three-time longitudinal study that analyzes the evolution of a large number of students during three semesters in higher education; (b) it mixes objective and subjective student data; (c) it tests the ECCT, the admissions test that was designed for this specific country and that has been previously validated; and, finally, (d) it uses a novel methodology and dominance analysis. This analysis has only been used occasionally in work and organizational psychology [46]. Although quite rarely applied, dominance analyses could provide interesting theoretical and practical insights into many questions in management, business, and work and educational psychology.

## 5. Conclusions

To date, there is no study that fully explains the explained variance in academic performance. There is a large body of literature that attempts to address the prediction of academic performance from various angles. This study shows how much variance in future academic performance is uniquely explained by cognitive and emotional aptitudes and past performance. Although past performance is the strongest predictor, each variable is necessary to understand academic performance. Even with its limitations (i.e., some of the percentages of our variables are very low; the sample belonged solely to one center in Latin American), this paper confirms the importance of cognitive aptitudes and past performance, giving exact values of their contribution to academic performance over time and opening a new path of research with emotional intelligence. Undoubtedly, more research is necessary to understand how emotional intelligence predicts academic performance.

## 6. Patents

The ECCT was patented in 2015 by Cortés, J.A. and Vera, M.

## Figures and Tables

**Figure 1 ijerph-18-13184-f001:**
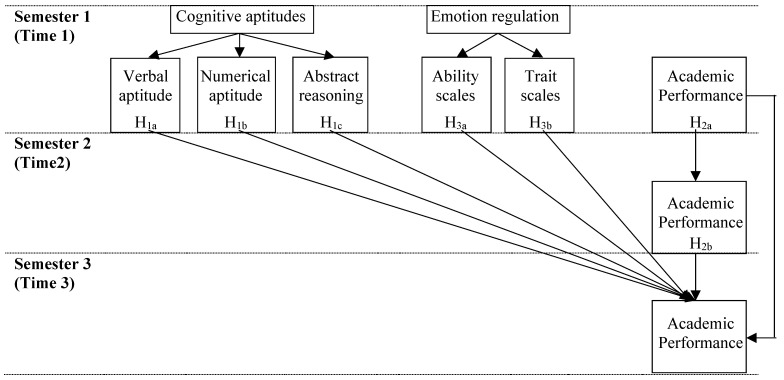
Variables under study and hypothesised model.

**Table 1 ijerph-18-13184-t001:** Descriptive statistics and correlations of variables in the study.

	Mean	SD	Int. cons.	CR	AVE	1	2	3	4	5	6	7
1. Verbal aptitude	20.88	3.45	0.57	0.72	0.35							
2. Numerical aptitude	22.54	5.11	0.83	0.85	0.66	0.39 ***						
3. Abstract reasoning	22.84	3.01	0.66	0.80	0.66	0.31 ***	0.42 ***					
4. Scenarios	14.65	2.70	0.87	0.91	0.62	0.28 ***	0.24 ***	0.17 ***				
5. Self-questionnaire	4.40	0.52	0.90	0.81	0.17	0.01	−0.05	−0.07 *	0.16 ***			
6. T1 Academic perf.	8.18	0.49				0.16 ***	0.23 ***	0.05	0.14 ***	0.08 *		
7. T2 Academic perf.	8.10	0.60				0.15 ***	0.14 ***	0.03	0.11 **	0.09 **	0.63 ***	
8. T3 Academic perf.	7.82	1.19				0.18 ***	0.17 ***	0.03	0.15 ***	0.12 ***	0.62 ***	0.69 ***

Note: * *p* < 0.05 ** *p* < 0.01; *** *p* < 0.001. Spearman correlations. Int. cons. = internal consistency through KR20 for variables 1–3 and Cronbach´s alpha for 4–5.

**Table 2 ijerph-18-13184-t002:** Gender among variables under study.

	Boys	Girls	z	*p*
Verbal aptitude	21.37	20.56	−2.92	0.004
Numerical aptitude	23.82	21.72	−5.62	0.000
Abstract reasoning	23.82	22.21	−7.86	0.000
Scenarios	14.52	14.74	−1.22	0.224
Self-questionnaire	4.28	4.48	−5.32	0.000
T1 Academic perf.	8.03	8.27	−6.87	0.000
T2 Academic perf.	7.93	8.20	−6.00	0.000
T3 Academic perf.	7.53	8.02	−8.82	0.000

Note: In boys and girls row, mean is showed. Z from Mann-Whitney test.

**Table 3 ijerph-18-13184-t003:** Degree among variables under study.

	Bank	Pharmacy	HHRR	Industrial	Tourism	System	Child	Graphic	Marketing	Optometry	χ^2^ _(9)_	*p*
Verbal aptitude	20.03	21.45	20.47	20.29	21.81	21.81	20.39	21.55	21.54	22.39	37.00	0.000
Numerical aptitude	23.16	20.50	20.81	23.71	22.59	25.73	21.34	23.45	21.03	24.86	75.88	0.000
Abstract reasoning	22.53	22.60	21.79	23.65	23.10	24.50	22.04	24.26	22.83	23.44	72.44	0.000
Scenarios	14.67	15.21	14.81	14.50	14.68	15.22	14.50	14.03	14.42	14.86	9.11	0.427
Self-questionnaire	4.36	4.47	4.44	4.31	4.50	4.34	4.49	4.20	4.55	4.44	23.64	0.005
T1 Academic perf.	8.20	7.83	8.43	8.27	8.14	7.72	8.33	7.98	8.50	7.89	179.23	0.000
T2 Academic perf.	8.15	7.75	8.44	8.25	8.72	7.44	7.95	7.73	8.61	7.82	282.93	0.000
T3 Academic perf.	7.78	8.06	8.03	7.83	8.18	7.03	8.01	7.35	8.10	7.78	154.34	0.000

Note: χ^2^ from Kruskal-Wallis test.

**Table 4 ijerph-18-13184-t004:** Results of general dominance analyses for T3 academic performance in the whole sample.

	Estimate	Rank	%
Verbal aptitude	0.004	6	1.54%
Numerical aptitude	0.008	4	3.08%
Abstract reasoning	0.003	7	1.15%
Scenarios	0.004	5	1.54%
Self-questionnaire	0.001	8	0.38%
T1 Academic perf.	0.095	2	36.54%
T2 Academic perf.	0.120	1	46.15%
Gender	0.023	3	8.85%
R^2^/% R^2^	0.258		100%

Note: Standardized dominance estimates, rank and explained variances for T3 academic performance.

**Table 5 ijerph-18-13184-t005:** Results of general dominance analyses for T3 academic performance separately by degree.

	Bank			Pharmacy			HHRR			Industrial			Tourism			System			Child			Graphic			Marketing			Optometry		
	E	R	%	E	R	%	E	R	%	E	R	%	E	R	%	E	R	%	E	R	%	E	R	%	E	R	%	E	R	%
Verbal	0.018	4	5.14%	0.040	3	12.74%	0.005	8	1.19%	0.017	5	2.90%	0.014	5	4.65%	0.049	3	19.44%	0.009	5	3.91%	0.009	4	4.43%	0.018	6	5.20%	0.024	6	3.50%
Numerical	0.051	3	14.57%	0.036	4	11.46%	0.014	4	3.34%	0.073	3	12.46%	0.010	6	3.32%	0.007	6	2.78%	0.013	4	5.65%	0.085	1	41.87%	0.030	4	8.67%	0.033	5	4.81%
Abstract	0.008	6	2.29%	0.012	6	3.82%	0.008	6	1.91%	0.012	7	2.05%	0.059	3	19.60%	0.001	8	0.40%	0.029	3	12.61%	0.006	6	2.96%	0.019	5	5.49%	0.011	8	1.60%
Scenarios	0.004	7	1.14%	0.022	5	7.01%	0.008	5	1.91%	0.012	6	2.05%	0.063	2	20.93%	0.005	7	1.98%	0.003	7	1.30%	0.001	8	0.49%	0.010	7	2.89%	0.312	1	45.48%
Self-quest	0.002	8	0.57%	0.007	7	2.23%	0.006	7	1.43%	0.004	8	0.68%	0.001	8	0.33%	0.012	4	4.76%	0.004	6	1.74%	0.006	5	2.96%	0.002	8	0.58%	0.012	7	1.75%
T1 A. perf.	0.116	2	33.14%	0.141	1	44.90%	0.179	1	42.72%	0.146	2	24.91%	0.051	4	16.94%	0.087	1	34.52%	0.045	2	19.57%	0.081	2	39.90%	0.069	3	19.94%	0.053	3	7.73%
T2 A. perf.	0.141	1	40.29%	0.052	2	16.56%	0.176	2	42.00%	0.237	1	40.44%	0.097	1	32.23%	0.081	2	32.14%	0.127	1	55.22%	0.013	3	6.40%	0.125	1	36.13%	0.194	2	28.28%
Gender	0.013	5	3.71%	0.005	8	1.59%	0.023	3	5.49%	0.034	4	5.80%	0.005	7	1.66%	0.010	5	3.97%	0.001	8	0.43%	0.002	7	0.99%	0.073	2	21.10%	0.046	4	6.71%
R^2^/% R^2^	0.353		100%	0.314		100%	0.419		100%	0.586		100%	0.301		100%	0.252		100%	0.230		100%	0.203		100%	0.346		100%	0.686		100%

Note: E = Standardized dominance estimates; R = rank; and % = explained variances for T3 academic performance.

## Data Availability

Data are available upon reasonable request to Vera, M.

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
