# Peer review of "Emotional and Cognitive Aptitudes and Successful Academic Performance: Using the ECCT"

_ijerph, 2021, doi:10.3390/ijerph182413184_

Round 1
Reviewer 1 Report
It is a well written article addressing a topic of great interest and controversy for various stakeholders, mainly universities, which is academic performance and its predictability.
However, why is it relevant to the section Digital Health of the International Journal of Environmental Research and Public Health ? Why not to Education Sciences? You state from the very beginning that “Understanding factors that influence academic performances is vital” but for what and for whom? You do briefly state in the Introduction that “Education is a key aspect of the development of any country.” And we might infer that your research could be relevant for the areas the IJERPH covers. Or maybe not?
There is a clear structure in which you critically engage with the materials you present. You offer a good review of the state of research for the various variables chosen for your research. The rationale for the research is explicit and plausible with a clear presentation of its weaknesses and strengths in the respective section (471 – 496).
There’s a formatting improvement that needs to be done in line 92.
Author Response
Thank you very much for your kind words. Regarding the section, the Journal has decided it, but I completely agree with you that it is more appropriate Education Sciences, thus, I will ask editor about it.
Finally, I have changed Amer-ica, for Ame-rica in line 92.
Thank you for your time and your suggestions. We appreciate it.
Reviewer 2 Report
It is indeed an interesting study to explore the different cognitive and emotional aptitudes and past academic performance as predictors of academic performance.
There are some general questions for the authors
It appears through the manuscript that the final sample is derived from the third semester and the participants belong to 10 different specialities. Any specific reason for the authors to choose only these specialties?
Have the authors considered the learning methods, delivery of the curriculum or the pedagogy, and learning resources also as parameters to correlate the academic performance? (For example, the role of synchronous and asynchronous learning and digital learning tools in academic performance)
Have the authors found any need changes for the students depending on the outcomes of the study?
Author Response
Regarding specialties, there has not been a selection, but all the existing ones in the education centre have been used. We make it clearer in the sample section:
This institute offers a total of ten degrees as educational offer (Pag. 4, line 199)
Concerning the learning methods, although it is a very interesting topic, we have not taken into account in this paper because all students received the same method (as they are in the same centre), thus, we could not compared it. But it is a great idea in future studies. Unfortunately the COVID pandemic has forced a change in the teaching methodology and in the near future, we may have data to include this variable in the model.
Finally, these results generated a deep thought, not on the individual students, but on the differences between the different degrees and therefore on the skills needed to succeed in each one. So we adapted the ECCT depending on the career chosen, weighing the competencies measured.
Thank you very much for your time, your comments, and specially your questions that have made us consider future studies. We appreciate it.
Reviewer 3 Report
Dear authors,
I consider that the work deals with an interesting topic and it is well raised, but it needs some improvements that I will comment on.
In the abstract, I do not know understand the meaning of Time 1, Time 2 and Time 3. It should be explained before putting these expressions.
I believe that a broader and more independent section of Introduction should be made. In it, a general vision of the subject should be collected, as well as the objective, the methodology with which the study is approached, the results obtained briefly explained, the contribution of the study in different areas (academic, management, public, etc. ), and end with a paragraph that reflects the structure of the work.
Then, section 2 would come, which would be the theoretical framework, which would include the explanation that you mention about the different variables and the presentation of the hypotheses. It would be convenient to make a graph, with the different variables, the relationships between them and the hypotheses raised.
The conclusions must be expanded. The limitations and future lines of research should be included in this section of conclusions.
The work has potential and I consider it to be an interesting topic for the review.
All the best,
Author Response
Please, find attach the response

Reviewer 4 Report
Thank you for the opportunity to review the manuscript entitled, "Emotional and Cognitive Aptitudes and successful Academic Performance: Using the ECCT”.
I believe this study investigated a topic of little relevance to the readers of “IJERPH”.
Traditionally, academic performance has been conceptualized based on the knowledge and skills that a student demonstrates in a subject, operationalized in a final grade that repressents performance and academic achievement. Cognitive variables such as intelligence, skills and prior knowledge, conative variables such as cognitive and learning styles, and affective variables such as motivation and personality have been considered the individual factors responsible for academic performance and, consequently, obtaining good grades. Research on academic performance has not only assessed personal or individual factors, it has also focused on the effects of contextual variables.
In general, this paper is well written and follows well accepted standards of academic writing. However, my opinion is that in the current form the manuscript does not meet my expectations for publication in the journal. The study not enough of an advance or of enough impact for the investigation about Academic Performance and lacks Sophisticated and/or ambitious a statistical analysis (t-Student, ANOVA, Dominance Analyses)
Participants:
The participants were not randomly assigned. The number of participants from a Technological Institute in Ecuador is not representative and directly affects external validity. The Technological Institute in Ecuador was chosen at random?
Instruments:
The reliability of The Emotional and Cognitive Competence Test (ECC) was evaluated using Cronbach’s alpha measure. Cronbach's Alpha is conditioned by the number of items and the number of alternative responses, it is necessary to use other alternative reliability indices, such as Composite Reliability (CR) and McDonald's Omega (Ω), which are calculated through factorial loads and are measured more accurate reliability. Furthermore, it is also necessary to estimate the convergent validity using the Extracted Mean Variance (AVE).
The instruments must always display two important qualities: reliability and validity. It is good practice to perform a confirmatory factor analysis. Is need to assess the validity of the constructs of the ECC (Confirmatory Factor Analysis: Relative Chi-Square, P; IFI; GFI; AGFI; CFI; RMSEA).
Statistical Analysis.
The t-Student test requires univariate normality (Test Kolmogorov-Smirnov).
The ANOVA analysis requires key assumptions: independent variables, univariate normality (Test Kolmogorov-Smirnov) and homoscedasticity, the assumption of equality of variances (Test Levene).
The Dominance Analyses assumes that there is little or no multicollinearity in the data.
Author Response
Please, find attach the response

Round 2
Reviewer 3 Report
Dear authors,
I can not see the tables, figures and cover letter. Could you please enclose them?
Best regards,
"Once the correct version of the manuscript has been reviewed, I give the go-ahead and consider that the article can be accepted."